# Effect of living arrangement on anthropometric traits in first-year university students from Canada: The GENEiUS study

**Tanmay Sharma[1], Christine Langlois[1], Rita E. Morassut[1], David Meyre**[1,2]*

**1** Department of Health Research Methods, Evidence, and Impact, McMaster University, Hamilton, Canada,
**2** Department of Pathology and Molecular Medicine, McMaster University, Hamilton, Canada

* meyred@mcmaster.ca

**Data Availability Statement:** The data underlying the results presented in the study are available in the supplementary material.

## Abstract

### Background

The transition to university often involves a change in living arrangement for many first-year students. While weight gain during first year of university has been well documented, Canadian literature on the impact of living arrangement within this context is limited. The objective of this investigation was to explore the effect of living arrangement on anthropometric traits in first-year university students from Ontario, Canada.

### Methods

244 first-year undergraduate students were followed longitudinally with data collected early in the academic year and towards the end of the year. Anthropometric parameters including weight, waist and hip circumference, body mass index (BMI), and waist-to-hip ratio (WHR) were examined. The Wilcoxon signed-rank test was used for pairwise comparison of traits from the beginning to end the year in the absence of adjustments. Additionally, linear regression models with covariate adjustments were used to investigate effect of the type of living arrangement (i.e. on-campus, off-campus, or family home) on the aforementioned traits.

### Results

In the overall sample, a significant weight increase of 1.55kg (95% CI: 1.24–1.86) was observed over the school year (p<0.001), which was also accompanied by significant gains in BMI, and waist and hip circumferences (p<0.001). At baseline, no significant differences were found between people living on-campus, off-campus, and at home with family. Stratified analysis of change by type of living arrangement indicated significant gains across all traits among students living on-campus (p<0.05), and significant gains in weight and BMI among students living at home with family. Additionally, a comparison between living arrangements revealed that students living on campus experienced significantly larger gains in weight and BMI compared to students living off-campus (p<0.05).

**Funding:** DM holds a Canada Research Chair in Genetics of Obesity. TS is supported by the Canadian Institutes of Health Research Canada Graduate Scholarship. The funders had no role in study design, data collection and analysis, decision to publish, or preparation of the manuscript.

**Competing interests:** The authors have declared that no competing interests exist.

## Conclusion

Our findings indicate that living arrangement is associated with different weight gain trajectories in first-year university students.

## Introduction

The rising prevalence of overweight and obesity in the Canadian population is a cause for concern. According to Statistics Canada, 26.8% of the Canadian population was affected by obesity in 2018. In North America, the greatest increase in the number of individuals with obesity has been among those aged 18 to 29, with the transition from adolescence to adulthood being implicated as a sensitive time for dramatic and inappropriate weight gain [1]. Young adults have also experienced the greatest increase in the incidence of overweight and obesity in recent years, compared to adults in other age groups [2, 3]. While education status is negatively correlated with body mass index (BMI) in the general population from high-income countries, young adults in higher education gain more weight and are more likely to develop obesity than those without university education in the United States [1, 4, 5]. An American study reported that 69% of university students experienced an increase in BMI between the beginning of their first year at university and the end of their second year [6]. In the university student population, obesity affects 14% of American undergraduate students [7]. The comorbidities of obesity include depression, sleep apnea, chronic back pain, osteoarthritis, gallbladder disease, type 2 diabetes, fatty liver, hypertension, cardiovascular disease, and some cancers [8, 9]. Adolescence and young adulthood may be critical periods for the development of obesity as elevated body mass index (BMI) during this time is associated with chronic obesity, higher morbidity, and premature mortality [10–15].

The "Freshman 15" is the belief that incoming university student gain 15lb (~6.8kg) during their freshman year, yet the evidence for this is limited [16]. Previous studies have found this to be an exaggeration, estimating an average body weight (BW) increase of 3 to 5 lbs (1.4 to 2.3 kg) [17–19]. These observed BW changes may reflect underlying modifications in environmental factors, lifestyle habits and other health-related behaviours during the transition from secondary school to university [20].

The transition to university often involves a change in living arrangement for many first-year students. While some incoming students can commute to university for classes while still living at home with their families, others who live relatively farther from the university have to relocate and find temporary accommodation closer to the university for the duration of their studies. Generally, most first-year year students who relocate choose between one of two main options: i) applying to live in university residence on campus ii) finding shared rental accommodation near the university campus. In most cases, these arrangements involve living away from family and living with other students. Some previous studies have suggested that living arrangement can have a significant impact on BW and BMI during first year of university [21–23]. However, the number of Canadian studies within this context is relatively limited [21–23]. Furthermore, previous Canadian studies have only examined the broader differences between 'on-campus' and 'off-campus' living arrangements, and have not explored specific differences in weight change between the three most common living arrangement options (on-campus, off-campus, or family home) available to first-year university students [21–23]. This prompted us to study the effect of living arrangement on five anthropometric traits in a multiethnic sample of 244 undergraduate students from McMaster University in Hamilton, Ontario (Canada).

## Participants and methods

### Participants

Genetic and EnviroNmental Effects on weight in University Students (GENEiUS) was a prospective observational study which investigated the environmental and biological determinants of obesity trait changes in Canadian undergraduate students [24]. As part of this study, undergraduate students from McMaster University (Hamilton, Ontario) were followed every six months over four years, beginning in September of their first year of study. First-year undergraduate students enrolled at McMaster University, between the ages of 17 and 25, were eligible to participate in the study and were primarily recruited via in-class advertising on the main university campus and social media promotion. Individuals who were pregnant, had previously given birth, or had a medical condition which could have impacted their BMI for a long period of time (e.g. bariatric surgery, immobilization from injury) were excluded from the study. Additional details regarding the GENEiUS study have been described previously [24]. Written informed consent was obtained directly from the participants. All methods and procedures for this study were in accordance with the Declaration of Helsinki principles and were reviewed and approved by the Hamilton Integrated Research Ethics Board (REB#0524).

### Data collection

Four cohorts of participants (2015–2016, 2016–2017, 2017–2018, 2018–2019) were followed longitudinally with data collected at two study visits:: one towards the beginning of their first-year (September/October) and one towards the end of their first-year (March/April). A total of 361 participants were enrolled in the study between 2015 and 2018, of which 245 (68%) completed one year of follow-up (i.e. completed the first baseline visit around September/October and a second follow-up visit in March/April) between 2016 and 2019. Only 244 participants were analyzed in this investigation (i.e. only those participants who reported their living/housing arrangement status at baseline). Data analyzed in this study included anthropometric data (i.e. BW, BMI, waist circumference (WC), hip circumference (HC), waist hip ratio (WHR)), and demographic characteristics (i.e. sex, age, ethnicity, living arrangement, type of undergraduate program). Trained research personnel performed all anthropometric measurements in duplicate to reduce intra-rater variability. Participants wore light clothing and removed shoes before being weighed. BW was measured to the nearest 0.1 kg using a digital scale (Seca, Hamburg, Germany). Height was measured to the nearest 0.1 cm using a portable stadiometer (Seca 225, Hamburg, Germany). WC was measured after a normal exhalation at the midpoint of the last palpable rib and the superior portion of the iliac crest to the nearest 0.1 cm, and HC was measured at the widest part of the buttocks to the nearest 0.1 cm using a stretch-resistant tape measure, as recommended by the World Health Organization (WHO) [25]. WHR was calculated as WC divided by HC. BMI ($kg/m^2$) was calculated by dividing weight by squared height. Demographics information was collected through online, self-reported questionnaires.

### Statistical methods

All statistical analyses were performed using IBM SPSS Version 25 statistical package. Descriptive analysis was carried out to assess the baseline distribution of traits within the study sample. Data for continuous variables have been reported using means and standard deviations while categorical data have been reported by counts and percentages. Anthropometric data at each time point were screened for potential outliers. Any identified outlying

data points were individually cross-checked to determine if they were true outliers, representing participants who truly fell outside the general distribution of our data, or if the outliers were a result of inaccuracies in measurement or data transcription. Data inaccuracies were corrected while all other outliers were left in the dataset. All data were assessed graphically and statistically for normality of distribution prior to analysis. The non-parametric Wilcoxon signed-rank test was used for pairwise comparison of anthropometric traits (i.e. BW, BMI, WC, HC, WHR) at baseline and after 6 months (i.e. beginning and end of the 1st year). The effect of living arrangement on anthropometric traits at baseline and change by the end of first year were analyzed using linear regression models with adjustment for covariates including sex, cohort of entry (i.e. 2015–2016, 2016–2017, 2017–2018, 2018–2019), and baseline trait values. A rank-based inverse normal transformation was applied to all outcome variables deviating from normality. In our analysis of association, we examined living arrangement in two different ways. Firstly, we examined living arrangement as a ternary variable with three different response categories (i.e. living in university residence, living in student rental housing off-campus, or living at home with family) to evaluate the specific differences between the three main living arrangements options available to most first-year university students. In addition to that, given that a relatively smaller proportion of students in our sample lived in either off-campus rental housing or at home with family, we combined these two categories into one, in order to increase the relative sample size for comparison, based on the fact that both these categories represent an 'off-campus' living environment. Subsequently, as part a secondary analysis, we examined living arrangement as a binary variable to investigate the differences between 'on-campus' and 'off-campus' living arrangements. Given that i) the present study is hypothesis-driven; ii) the research questions have been previously tested in literature; iii) the tested obesity outcomes are not independent, a Bonferroni correction was not applied, as even though it reduces the chance of making type I errors, it can increase the chance of making type II errors [26, 27]. Therefore, the level of statistical significance was set at p <0.05 for all tests.

## Results

### Participant characteristics

Three hundred and sixty one participants were enrolled into the study of which 245 (68%) completed one year of follow up. Only 244 participants were analyzed in this investigation (i.e. only those participants who reported their living arrangement at baseline). The average follow up time between the baseline visit, early in the academic year, and the follow up visit, towards the end of the year, was 21.6 (SD = 2.18) weeks. Participants displayed an average age of 17.87 (SD = 0.48) years. Female participants represented 80.7% of the sample (n = 197). In terms of living arrangement, at baseline 69.7% of the participants reported living in university residence on campus (n = 170), 19.7% reported living at home with family (n = 48), and 10.7% reported living away from family in off-campus student rental housing (n = 26). Participants of East Asian ethnicity represented 31.1% of the sample (n = 76), white-Caucasian participants represented 25% (n = 61), participants of South Asian ethnicity represented 18.4% (n = 45), participants with mixed ethnic background represented 12.7% (n = 31), participants with Middle Eastern background represented 7% (n = 17), and participants belonging to other ethnic groups including African, Latin American, Pacific Islander, and Canadian Indigenous collectively represented 5.7% (n = 14) of the sample. Out of the participants who reported their program of study, 86.1% reported being enrolled in a science-based academic program (e.g. Health Science, Life Science, Kinesiology, Engineering), while 13.9% reported being enrolled in a non-science academic program (e.g. Humanities, Business, Arts).

**Table 1. Overall anthropometric trait trends in first year of university.**

|  | Beginning | End | Change | P-value* |
|---|---|---|---|---|
|  | Mean (SD) | Mean (SD) | MD (95% CI) |  |
| **Body Weight (kg)** | 60.43 (12.00) | 61.98 (12.41) | 1.55 (1.24–1.86) | **<0.001** |
| **BMI (kg/m$^2$)** | 21.52 (3.35) | 22.17 (3.46) | 0.65 (0.53–0.77) | **<0.001** |
| **Waist Circumference (cm)** | 75.12 (8.69) | 76.26 (9.00) | 1.14 (0.63–1.66) | **<0.001** |
| **Hip Circumference (cm)** | 97.19 (7.75) | 98.12 (7.45) | 0.93 (0.55–1.31) | **<0.001** |
| **WHR** | 0.772 (0.049) | 0.776 (0.054) | 0.004 (-0.001–0.009) | 0.086 |

Data are expressed as mean (SD) and mean difference (95% CI); WC data not collected for one participant; Abbreviations: BMI, body mass index; WHR, Waist to hip ratio; MD, Mean difference.

*Non-parametric pairwise comparison (non-adjusted comparison of change in outcomes from beginning to end of school year). P-values below 0.05 represented in bold font.

## Overall trends in anthropometric traits in first year of university

At baseline, the mean BW, BMI, WC, HC, and WHR for the overall sample was 60.43kg (SD = 12.00), 21.52 kg/m$^2$ (SD = 3.35), 75.12cm (SD = 8.69), 97.19cm (SD = 7.75), and 0.772 (SD = 0.049) respectively. In terms of their weight status category at baseline, 78.3% (n = 191) of the participants were normal weight (18.5 kg/m$^2$ ≤ BMI < 25 kg/m$^2$), 12.3% (n = 30) were underweight (BMI < 18.5 kg/m$^2$), 6.6% (n = 16) were overweight ((25 kg/m$^2$ ≤ BMI < 30 kg/m$^2$)), and 2.9% (n = 7) were obese (BMI ≥ 30 kg/m$^2$). By the end of the academic year, significant increases in BW (1.55 kg, 95% CI: 1.24–1.86; p<0.001), BMI (0.65 kg/m$^2$, 95% CI: 0.53–0.77; p<0.001), WC (1.14 cm, 95% CI: 0.63–1.66; p<0.001), and HC (0.93 cm, 95% CI: 0.55–1.31; p<0.001), but not WHR (0.004, 95% CI: -0.001–0.009; p = 0.086), were noted in the overall sample, when compared to baseline. Table 1 summarizes the aggregated data at each time point for all investigated anthropometric traits. The average rate of weight change over the academic year was +0.072 kg/week (SD = 0.12). Some previous studies have deemed a 5% change in overall body weight to be clinically meaningful with a weight loss of at least 5% being associated with improvements in blood pressure, HDL cholesterol, depression, and overall quality of life as well as reduction in health care costs [28]. In our sample, 28.7% (n = 70) of the participants experienced clinically meaningful weight gain, based on the 5% weight change threshold, with the average weight gain among these participants being 3.65 (SD = 3.02) kg. Lastly, in our overall sample, only 2.9% (n = 7) of the participants gained 15 pounds or more, as predicted by the popularized theory of 'Freshman 15'. In terms of their weight status category by the end of the academic year, 77.0% (n = 188) of the participants were normal weight, 8.6% were underweight (n = 21), 11.5% were overweight (n = 28), and 2.9% (n = 7) were obese. More information (box plots, histograms) regarding the distribution of the five anthropometric traits (BW, BMI, WC, HC, and WHR) at the beginning and the end of the first year of university is available in the supplementary online material (S1–S20 Figs).

## Trends in anthropometric traits based on living arrangement in first year of university

At the beginning of the academic year, there were no significant baseline differences in BW, BMI, WC, HC, and WHR between participants living in university residence on campus, those living away from family in off-campus student rental housing, and those living at home with family (p ≥ 0.05 for all comparisons). When examining change from the beginning to the end of the academic year, significant increases were noted across all investigated traits among

students living in university residence (p < 0.05 for all traits). In comparison, while students living away from family in off-campus housing displayed modest gains over the year across all parameters, none reached the threshold of statistical significance (p ≥ 0.05 for all traits). Among students living at home with family, significant changes were observed in only BW and BMI by the end of the year relative to baseline (p<0.05 for both traits). Table 2 presents the trends in the investigated anthropometric traits from the beginning to the end of the academic year categorized by living arrangement.

Table 3 compares the differences in anthropometric parameters at baseline and change over first year of university between the different living arrangements. In this case, we analyzed living arrangement as a ternary variable, to evaluate the differences between the three specific types of living arrangements, as well as a binary variable to evaluate overall differences between 'on-campus' and 'off-campus' living arrangements. Given that a relatively lower number of participants in our sample lived in either off-campus student housing (n = 26) or at home with family (n = 48), we combined these two 'off-campus' living arrangements into one mutual

**Table 2. Trends from beginning to the end of first year in students living on campus residence (n = 170), in off-campus housing (n = 26), and at home with family (n = 48).**

| | Living Arrangement | Beginning | End | Change | P-value[*] |
|---|---|---|---|---|---|
| | | Mean (SD) | Mean (SD) | MD (95% CI) | |
| **Body Weight (kg)** | University Residence | 60.27 (11.48) | 62.13 (11.91) | 1.86 (1.52–2.20) | **<0.001** |
| | Off-Campus Student Housing | 60.92 (13.42) | 61.53 (13.42) | 0.61 (-0.51–1.72) | 0.204 |
| | At home with family | 60.71 (13.24) | 61.67 (13.78) | 0.96 (0.16–1.76) | **0.017** |
| **BMI (kg/m$^2$)** | University Residence | 21.44 (3.10) | 22.20 (3.28) | 0.76 (0.63–0.89) | **<0.001** |
| | Off-Campus Student Housing | 21.65 (4.03) | 21.91 (3.69) | 0.26 (-0.17–0.70) | 0.144 |
| | At home with family | 21.72 (3.85) | 22.17 (4.00) | 0.45 (0.14–0.76) | **0.005** |
| **Waist Circumference (cm)** | University Residence | 74.57 (7.91) | 76.07 (8.57) | 1.50 (0.95–2.06) | **<0.001** |
| | Off-Campus Student Housing | 75.83 (9.82) | 77.17 (9.68) | 1.34 (-0.44–3.13) | 0.098 |
| | At home with family | 76.69 (10.55) | 76.47 (10.25) | -0.23 (-1.70–1.24) | 0.739 |
| **Hip Circumference (cm)** | University Residence | 96.98 (7.23) | 98.02 (7.13) | 1.04 (0.59–1.50) | **<0.001** |
| | Off-Campus Student Housing | 97.83 (9.14) | 98.38 (8.84) | 0.55 (-0.32–1.42) | 0.204 |
| | At home with family | 97.59 (8.78) | 98.35 (7.93) | 0.75 (-0.23–1.73) | 0.181 |
| **WHR** | University Residence | 0.768 (0.048) | 0.775 (0.054) | 0.007 (0.001–0.013) | **0.014** |
| | Off-Campus Student Housing | 0.775 (0.052) | 0.784 (0.053) | 0.009 (-0.007–0.025) | 0.276 |
| | At home with family | 0.784 (0.052) | 0.775 (0.053) | -0.009 (-0.021–004) | 0.182 |

Data are expressed as mean (SD) and mean difference (95% CI); WC data not collected for one participant living in off-campus student housing; Abbreviations: BMI, body mass index; WHR, Waist to hip ratio; MD, Mean difference.

[*]Non-parametric pairwise comparison by living arrangement subgroups (non-adjusted comparison of change in outcomes from beginning to end of school year). P-values below 0.05 represented in bold font).

**Table 3. Association between living arrangement and anthropometric traits in first year of university.**

| | | Living Arrangement Categorical | | | Living Arrangement Binary |
|---|---|---|---|---|---|
| | | University Residence *vs.* Home with Family[5] | Off-Campus Student Housing *vs.* Home with Family[5] | University Residence *vs.* Off-Campus Student Housing[5] | Living On-Campus *vs.* Living Off-Campus[5] |
| Body Weight (kg) | Baseline[1] | -0.037 (0.142), 0.793 | 0.027 (0.212), 0.897 | -0.065 (0.184), 0.726 | -0.047 (0.121), 0.700 |
| | Change[2] | 0.946 (0.399), **0.019** | -0.272 (0.597), 0.649 | 1.218 (0.518), **0.020** | 1.040 (0.341), **0.003** |
| BMI (kg/m$^2$) | Baseline[1] | -0.041 (0.159), 0.798 | -0.044 (0.238), 0.852 | 0.004 (0.206), 0.986 | -0.025 (0.136), 0.852 |
| | Change[2] | 0.367 (0.149), **0.015** | -0.159 (0.223), 0.475 | 0.526 (0.193), **0.007** | 0.422 (0.127), **0.001** |
| Waist Circumference (cm) | Baseline[1] | -0.163 (0.150), 0.278 | -0.128 (0.224), 0.567 | -0.035 (0.194), 0.859 | -0.118 (0.128), 0.356 |
| | Change[3] | 0.328 (0.163), **0.046** | 0.354 (0.246), 0.151 | -0.027 (0.215), 0.901 | 0.208 (0.141), 0.141 |
| Hip Circumference (cm) | Baseline[1] | -0.102 (0.156), 0.515 | -0.025 (0.233), 0.915 | -0.077 (0.202), 0.705 | -0.093 (0.133), 0.486 |
| | Change[3] | 0.049 (0.159), 0.759 | -0.033 (0.237), 0.891 | 0.081 (0.206), 0.693 | 0.060 (0.136), 0.658 |
| WHR | Baseline[1] | -0.276 (0.146), 0.060 | -0.255 (0.218), 0.244 | -0.021 (0.190), 0.913 | -0.187 (0.125), 0.136 |
| | Change[4] | 0.008 (0.006), 0.149 | 0.016 (0.009), 0.060 | -0.008 (0.008), 0.300 | 0.003 (0.005), 0.561 |

[1]Linear regression with rank-based inverse normal transformation, adjusted for sex and cohort;

[2]Linear regression adjusted for sex, baseline values, and cohort;

[3]Linear regression with rank based inverse normal transformation, adjusted for sex, baseline values, and cohort;

[4]Linear regression adjusted for sex, cohort, baseline WHR, baseline BMI, and BMI Change;

[5]β (Std. Error), p-value. Abbreviations: BMI, body mass index; WHR, Waist to hip ratio.

category (n = 74) to boost the sample size for comparison, and subsequently evaluated the overall differences between 'on-campus' and 'off-campus' living environments.

When considering the specific type of living arrangement (i.e. university residence, off-campus student housing, family home), living in university residence was found to be significantly associated with larger changes in BW (1.86 kg, 95% CI: 1.52–2.20) and BMI (0.76 kg/m$^2$, 95% CI: 0.63–0.89), after adjustment for sex, cohort, and baseline values, when compared to living at home with family [BW: 0.96 kg (95% CI: 0.16–1.76), P = 0.019; BMI: 0.45 kg/m$^2$ (95% CI: 0.14–0.76, P = 0.015], and living in off-campus student rental housing [BW: 0.61 kg (95% CI: -0.51–1.72), P = 0.020; BMI: 0.26 kg/m$^2$ (95% CI: -0.17–0.70), P = 0.007]. Interestingly, in this case, the observed weight gain among students living in university residence was almost two times as much as students living at home with family (1.86 kg vs. 0.96 kg), and almost three times as much as students living in off-campus housing away from family (1.86 kg vs. 0.61 kg). This trend was also observed with respect to the observed change in BMI. In contrast, living in university residence was not significantly associated with increased changes in HC and WHR, when compared to both living at home and living in off-campus student housing. With respect to change in WC, while a significant association was noted when comparing living in university residence to living at-home (p = 0.046), no significant difference was found between the former and off-campus student housing. Lastly, when comparing off-campus student housing to at-home living, no significant differences in change were noted for any of the investigated anthropometric parameters.

When considering binary living arrangement status (i.e. living on-campus *vs.* off-campus), living on-campus was significantly associated with increased change in BW and BMI, with adjustment for sex, cohort, and baseline values, when compared to living off campus [BW: 1.86 kg (95% CI: 1.52–2.20) *vs.* 0.83 kg (95% CI: 0.20–1.47), P = 0.003; BMI: 0.76 kg/m$^2$ (95% CI: 0.63–0.89) *vs.* 0.38 kg/m$^2$ (95% CI: 0.14–0.63), p = 0.001). In comparison, there was no significant association found between living on-campus and change in WC, HC, and WHR relative to living off campus. Notably, in this case, students living on-campus gained approximately twice as much weight and BMI as students living off-campus.

## Discussion

In this investigation, we examined the effect of living arrangement on obesity related anthropometric traits in first year of university. The investigation brought forth several important results. In terms of overall sample trends, our results suggest that, on average, first-year students experience significant gains in BW, BMI, WC, and HC, but not WHR, by the end of the school year compared to early on in the year. When examining specific trends by living arrangement, we found no significant differences at baseline between the participants living in the three different types of living arrangements for any of the investigated traits. However, when examining the patterns of change within the three separate living arrangement subgroups, we found that only the students living in university residence displayed significant gains across all five investigated traits by the end of the academic year relative to baseline. In comparison, students living at home with family displayed significant gains in only BW and BMI, while students living in off-campus student housing displayed no significant changes over the academic year in any of the investigated traits. Lastly, when comparing the change observed between the different types of living arrangements, we found that living in university residence was associated with an increased change in BW and BMI, when compared to living in either type of off-campus living arrangement, and an increased change in WC when compared to only living at home with family. Notably, our data suggests that first-year students living in university residence gain approximately twice as much weight and BMI as students living at home, and almost thrice as much weight and BMI as students living in off-campus student rental housing. This pattern was consistent when examining living arrangement as a binary factor wherein students living on-campus gained approximately twice as much weight as students living off-campus.

In terms of general trends, an average weight gain of 1.55kg (3.4 pounds) was noted in our overall sample. While the observed overall mean weight change in our sample is modest compared to the popularized estimate of 15-pound (6.8 kg) in the media, our result is comparable to the overall pooled estimates of 1.36 kg and 1.75kg previously reported by Vadeboncoeur *et al.* [18], and Vella-Zarb and Elgar [19] respectively.

When examining living arrangement options among first-year university students in Canada, it is important to understand the underlying context in terms of how students choose which universities to apply to, and how that ultimately affects their choice of living arrangement. Generally, for many students in Canada, going to university away from home in another city or sometimes even in another province, is not an uncommon practice. There are different potential reasons for this. One of the contributing factors is the geographic location. Considering the province of Ontario as a case in point, there are a total of 21 recognized universities in the province currently with campuses in only 30 communities. Hence, many students who live far from these locations have to travel or relocate temporarily to pursue post-secondary education. Another important factor that often plays a role is student preferences regarding undergraduate programs or institutions. In some cases, certain programs are only offered by certain universities, or alternatively, students see more value in enrolling at a particular university based on their educational goals and the opportunities available at that institution. This is an important consideration for many students and influences which universities and programs students choose to apply to for their undergraduate studies. Last but not least, the university admission process is a selective one in Canada, so students are not automatically accepted to a university that is closer to their home. While the aforementioned factors play a role in the choices that many students make within this context, it is important note that these decisions can be further influenced by additional factors such as socioeconomic status, accessibility, and family needs. Nevertheless, these reasons partly explain why Canadian students tend to apply

to different universities, whether close to home or farther away, and that decision ultimately influences their choice of living arrangement.

For students who can attend a post-secondary institution either in their hometown or relatively close to their hometown, living with family and commuting to school from home is an accessible option. However, for students who are originally from places that are farther away from the university, such as international students, out of province students, or even students living in cities far from the institution, relocating to a place that is closer to the university campus is the only viable option. In such cases, many incoming first-year students may prefer living in university residence for a number of potential reasons. Given their lack of familiarity with the university lifestyle, the local surroundings, and in some cases even the local culture (for international students), university residence can be a relatively secured option for incoming first-year students as it entails a large number of resources and supports that may not be easily available or accessible outside the campus environment. Students living in university residence benefit from the convenience of living on campus where they are in close proximity to other first-year students, to their classes, and to a range of additional facilities such as the school cafeterias and other resource centres. Some previous Canadian reports within this context have indicated that living in residence during first year can help students develop new friendships and can also have a positive impact on overall academic outcomes [29, 30]. Notably, however, while the cost of living in university residence varies across Canada, it is generally a relatively expensive option.

Among students who relocate for their university education, some choose to live in shared renal accommodation in close proximity to the university campus. However, this is a relatively less common choice among incoming first-year students and/or their families as it does not involve the supports and resources that are usually available in university housing. This trend was also observed in our sample wherein the proportion of students living away from family in off-campus housing was the smallest out of the three living arrangement options. Nevertheless, there are different potential reasons as to why students may choose this living arrangement. Firstly, given that living in residence is relatively expensive, there may be a financial consideration for some of the students who opt for off-campus housing, which can be a relatively cheaper option as rent and other expenses can be shared with roommates to lower cost. Alternatively, in some cases, students who relocate do not necessarily have a choice because of the lack of available spots in university residence, while sometimes it is simply a matter of personal preference. Altogether, these are some of the factors that generally influence the choice of living arrangement in first-year of university, and also potentially explains the disproportionality observed in the distribution of students across the three major living arrangement options.

When examining change in anthropometric traits by living arrangement, we found that students living on-campus displayed significant gains across all investigated traits over the academic year, and exhibited significantly higher gains in BW and BMI compared to students in either type of off-campus living arrangement. These results have important implications as they indicate that first-year students are not all equally prone to weight gain and that instead susceptibility may vary based on the type of living arrangement.

Our findings are consistent with prior Canadian reports within this context. For instance, in a previous investigation, Vella-Zarb and Elgar [23] found that students living on campus gain significantly more weight than students living off-campus. Similarly, Pliner and Saunders [21] found that students living on campus, and particularly those with restrained eating patterns, experience larger gains in BMI than their counterparts living at home, while Provencher et al. [22] found a significant difference in weight change between male students living in residence and those commuting from home. There are several possible explanations for the observed results. Living in university residence has been associated with increased accessibility

to food, increased food storage within student dormitory rooms, lack of healthy food options, and overall unhealthy eating patterns [31–34]. Additionally, in several Canadian universities, purchasing a meal plan is compulsory when living in residence. This requirement mandates students to set aside a certain amount of money at the beginning of the year that can be subsequently used for purchasing food on campus during the year. In some cases, this money cannot roll over to the next year or be transferred back to the students. As such, students are sometimes compelled to purchase food excessively in order to use up all their meal plan money by the end of the school year, and hence consume more than they may otherwise. Altogether, we postulate that a combination of these aforementioned factors pertaining to unhealthy food choices on campus and increased food consumption due to mandated meal plans may be critical contributors to the weight gain observed among students living in residence [35]. This may not be surprising as, even in the general population, excessive energy intake has been considered to be one of the primary drivers of the current obesity pandemic [36, 37]. Interestingly, however, in their investigation, Pliner and Saunders [21] found that students in university residence with a restrained dietary regime gain the most weight. This finding is paradoxical and ultimately highlights the need to further explore specific eating behaviors within this population to understand how eating patterns in university explain the change in anthropometric traits during first year of university. Furthermore, it also suggests that the effect of living arrangement on anthropometric traits may be influenced by additional variables. For instance, when considering other contributing factors within this context, previous studies have also found that having roommates or living with peers can influence different health related behaviors including the choice of meal plan, smoking, alcohol consumption, and overall tendency to lose or maintain weight [38–41]. Ultimately, this collectively highlights some of the factors that potentially make the residence living environment more obesogenic than the off-campus alternatives, and may potentially explain why first-year students living on-campus in university residence gain more weight than their counterparts in living off-campus.

When considering students living off-campus, a significant increase in BW and BMI was observed over the academic year among students living at home with family. There may be a few different explanations for this observed trend. Firstly, students who live at home and commute to university on a daily basis for classes tend to typically spend a large amount of their time on campus, as they do not prefer going back home during breaks between classes due to the extensive commute time. As such, we postulate that while these students spend a large amount of time on campus, many of them purchase food on campus where unhealthier food options are more accessible. When considering cost of purchasing food, it is commonly known that healthier food options are more expensive. A systematic review and meta-analysis by Rao et al. [42] found that eating healthy can cost up to $1.50 more per person per day than eating unhealthy. Hence, in such cases, some students may be more likely to consume fattening foods as compared to healthier foods, which may partly explain the increase in weight and BMI observed in this group. Notably, however, in comparison to students living on campus, students living at home with family do not solely rely on food from cafeterias on campus and have the opportunity to consume more home-cooked meals, which may partly explain why they do not gain as much weight as students living in residence. Apart from that, increased commuting time may be an additional factor that contributes to the significant weight gain among students living at home. For many students who commute from home, the commute time can be upwards of an hour. This can have a significant impact on their physical and mental health and ultimately influence their weight as commuting time has been linked with decreased levels of life satisfaction, decreased physical activity, decreased sleep quality, and increased overall stress and fatigue [43, 44].

Lastly, in our sample, the students living away from family in off-campus housing displayed no significant changes over the academic year in any of the investigated traits. One potential explanation for this may be that students living independently in off-campus accommodation are more likely to cook at home, as supposed to purchasing food from campus regularly. Additionally, many students living in such independent off-campus arrangements have a higher degree of active commute as many of them generally commute to campus by walking or bicycling. This can have a considerable impact as active commuting has been shown to be associated with decreased BMI and decreased odds of being obese or overweight [45, 46].

Strengths of this study include a longitudinal study design and investigation of multiple obesity related anthropometric traits. Additionally, to the best of our knowledge, this is the first Canadian study to comprehensively investigate the effect of the three most common types of living arrangements in first-year of university on a wide variety of anthropometric traits within this context. Our study also has several limitations. Firstly, our sample size was modest (N = 244) and hence was insufficiently powered to detect subtle effects. Additionally, our sample only included 26 participants who lived in off-campus student housing and as such the sample size for this group also may have been insufficient to reliably draw inferences. However, in our investigation, we included comparative analysis of on-campus versus off-campus living environments, which combined the two groups living off-campus, and hence provided a relatively lager sample size for comparison. Apart from that, we recognize that we did not account for physical activity as a covariate, and also did not account for potential changes in living arrangement between the two assessment time points. As such, our findings should be interpreted in light of these limitations. Lastly, our study was limited by a relatively high attrition rate which may have potentially biased the results.

In conclusion, our data provides support for the trend of weight gain among first-year university students from Ontario, Canada, and further implicates the type of living arrangement as an important predictor within this context. Ultimately, these results suggest that being in a particular living arrangement influences susceptibility to weight gain in first year of university, and highlight the need of taking living arrangement into consideration for prevention and mitigation efforts. These findings may also be critical in prompting further research in this area to understand the underlying factors that make certain living arrangements more obesogenic than others. Community based interventions in university residence have been previously shown to be effective in promoting physical activity and fruit and vegetable consumption among residents [47]. As such, given that increased BMI during young adulthood has been linked to chronic obesity later in life, understanding the predictors of weight gain in young adults at university may be a critical step forward towards effective prevention of obesity in the next generation. Further large-scale studies should be conducted to confirm these findings.

## Supporting information

**S1 Fig. Distribution of body weight (BW) at the beginning and end of first year among students living on campus residence (n = 170), in off-campus housing (n = 26), and at home with family (n = 48).**
(DOCX)

**S2 Fig. Distribution of body mass index (BMI) at the beginning and end of first year among students living on campus residence (n = 170), in off-campus housing (n = 26), and at home with family (n = 48).**
(DOCX)

**S3 Fig. Distribution of waist circumference (WC) at the beginning and end of first year among students living on campus residence (n = 170), in off-campus housing (n = 26), and at home with family (n = 48).**
(DOCX)

**S4 Fig. Distribution of hip circumference (HC) at the beginning and end of first year among students living on campus residence (n = 170), in off-campus housing (n = 26), and at home with family (n = 48).**
(DOCX)

**S5 Fig. Distribution of waist to hip ratio (WHR) at the beginning and end of first year among students living on campus residence (n = 170), in off-campus housing (n = 26), and at home with family (n = 48).**
(DOCX)

**S6 Fig. Distribution of weight change observed over the academic year among participants living in student residence on campus.**
(DOCX)

**S7 Fig. Distribution of body mass index (BMI) change observed over the academic year among participants living in student residence on campus.**
(DOCX)

**S8 Fig. Distribution of waist circumference (WC) change observed over the academic year among participants living in student residence on campus.**
(DOCX)

**S9 Fig. Distribution of hip circumference (HC) change observed over the academic year among student participants living in student residence on campus.**
(DOCX)

**S10 Fig. Distribution of waist to hip ratio (WHR) change observed over the academic year among student participants living in student residence on campus.**
(DOCX)

**S11 Fig. Distribution of weight change observed over the academic year among student participants living in off-campus housing.**
(DOCX)

**S12 Fig. Distribution of BMI change observed over the academic year among student participants living in off-campus housing.**
(DOCX)

**S13 Fig. Distribution of WC change observed over the academic year among student participants living in off-campus housing.**
(DOCX)

**S14 Fig. Distribution of HC change observed over the academic year among student participants living in off-campus housing.**
(DOCX)

**S15 Fig. Distribution of WHR change observed over the academic year among student participants living in off-campus housing.**
(DOCX)

**S16 Fig. Distribution of weight change observed over the academic year among student participants living at home with family.**
(DOCX)

**S17 Fig. Distribution of BMI change observed over the academic year among student participants living at home with family.**
(DOCX)

**S18 Fig. Distribution of WC change observed over the academic year among student participants living at home with family.**
(DOCX)

**S19 Fig. Distribution of HC change observed over the academic year among student participants living at home with family.**
(DOCX)

**S20 Fig. Distribution of WHR change observed over the academic year among student participants living at home with family.**
(DOCX)

**S1 Dataset.**
(XLSX)

## Acknowledgments

We are indebted to all participants of this study. We would also like to extend our thanks to Anika Shah, Roshan Ahmad, Baanu Manoharan, Adrian Santhakumar, Kelly Zhu, Guneet Sandhu, Dea Sulaj, Tina Khordehi, Ansha Suleman, Heba Shahaed, Andrew Ng, Tania Mani, Sriyathavan Srichandramohan, Deven Deonarain, Celine Keomany, Omaike Sikder, Isis Lunsky, Gurudutt Kamath, and Christy Yu for helping with data collection.

## Author Contributions

**Conceptualization:** David Meyre.

**Data curation:** Tanmay Sharma.

**Formal analysis:** Tanmay Sharma, David Meyre.

**Funding acquisition:** David Meyre.

**Investigation:** Tanmay Sharma, Christine Langlois, Rita E. Morassut, David Meyre.

**Methodology:** Tanmay Sharma, David Meyre.

**Project administration:** David Meyre.

**Supervision:** David Meyre.

**Validation:** David Meyre.

**Writing – original draft:** Tanmay Sharma, David Meyre.

**Writing – review & editing:** Christine Langlois, Rita E. Morassut.

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
