## [Decision Letter · Decision Letter 0]

3 Sep 2020

PONE-D-20-20895

Effect of living arrangement on obesity traits in first-year university students from Canada : the GENEiUS study

PLOS ONE

Dear Dr. Meyre,

Thank you for submitting your manuscript to PLOS ONE. After careful consideration, we feel that it has merit but does not fully meet PLOS ONE’s publication criteria as it currently stands. Therefore, we invite you to submit a revised version of the manuscript that addresses the points raised during the review process.

ACADEMIC EDITOR: Kindly recheck statistical analysis and revise the study title.

We look forward to receiving your revised manuscript.

Kind regards,

Rasheed Ahmad, Ph.D.

Academic Editor

PLOS ONE

Journal Requirements:

3.We note that you have indicated that data from this study are available upon request. PLOS only allows data to be available upon request if there are legal or ethical restrictions on sharing data publicly. For information on unacceptable data access restrictions, please see http://journals.plos.org/plosone/s/data-availability#loc-unacceptable-data-access-restrictions.

Reviewers' comments:

Reviewer's Responses to Questions

**Comments to the Author**

1. Is the manuscript technically sound, and do the data support the conclusions?

Reviewer #1: Partly

2. Has the statistical analysis been performed appropriately and rigorously? 

Reviewer #1: No

3. Have the authors made all data underlying the findings in their manuscript fully available?

Reviewer #1: Yes

4. Is the manuscript presented in an intelligible fashion and written in standard English?

Reviewer #1: Yes

5. Review Comments to the Author

Reviewer #1: In this study, Sharma and collaborators assessed weight gain trends in first year university students and compared the effects of living arrangements. The authors observed that students gained weight in their first year, and that there may be an additional effect of living on campus. They concluded that living arrangement is associated with different weight gain trajectories. New investigations in this area are of general interest to the field. However, there are several limitations that make the results difficult to interpret confidently. Comments are included below with the hope of aiding the authors in improving the quality of the work:

Comments

1. The title and text throughout refer to the measurements as ‘obesity parameters’ or ‘obesity traits’, but these parameters (weight, BMI, etc.) are not strictly descriptive of obesity. None of the subjects were considered obese or overweight (using reported BMIs). This makes the repeated references to obesity inappropriate and somewhat misleading.

2. A large proportion of the academic year falls during winter months. Is reduced activity a potential confounder?

3. Students were surveyed about their living arrangements at baseline, were there follow-up surveys to determine if their living arrangements had changed during the academic year?

4. Critical re-evaluation of the statistical analysis is warranted. The SDs are larger than the means in every case for the change data in table 2. Though a distribution-free test was used for comparison, it is still difficult to interpret the overall changes. Graphical representation of individual data points may be more informative. It is not clear how the distributions deviate from normality (e.g. skewed by weight loss, or excessive individual weight gain, etc)? Use of means for central tendency in skewed data is inappropriate. In the final discussion, only the group means are used to summarize support for the conclusions, but the individual changes are more relevant.

6. PLOS authors have the option to publish the peer review history of their article (what does this mean?). If published, this will include your full peer review and any attached files.

Reviewer #1: No

---

## [Author Response · Author response to Decision Letter 0]

17 Oct 2020

We would like to thank the editor and the reviewer for their exceptional input and suggestions on the article. We have addressed their comments to the best of our ability and we think that the revised version of the manuscript has significantly improved. 

Reviewer #1: 

1. The title and text throughout refer to the measurements as ‘obesity parameters’ or ‘obesity traits’, but these parameters (weight, BMI, etc.) are not strictly descriptive of obesity. None of the subjects were considered obese or overweight (using reported BMIs). This makes the repeated references to obesity inappropriate and somewhat misleading.

We thank the reviewer for this comment. The reviewer is correct that a minority of students reached the BMI cut-offs for overweight and obesity in this study. We have now changed ‘obesity parameters/traits’ to ‘anthropometric parameters or traits’ in the title and throughout the text to avoid this confusion. 

2. A large proportion of the academic year falls during winter months. Is reduced activity a potential confounder?

The reviewer brings up a valid point. Unfortunately, in this case, we did not account for physical activity as a covariate in our study because the parameter was not measured adequately in the entire cohort. We have added this point as one of the limitations of our study in the discussion section of the paper. 

“Apart from that, we recognize that we did not account for physical activity as a covariate, and did not account for potential changes in living arrangement between the two assessment time points. As such, our findings should be interpreted in light of these limitations.”

3. Students were surveyed about their living arrangements at baseline, were there follow-up surveys to determine if their living arrangements had changed during the academic year?

Thank you for the comment. Unfortunately, in this case we do not have that data. We agree with the reviewer that there is a possibility that a participant’s living arrangement could have changed over the year. However, in practice, we have noticed that students usually do not change their place of residence in the middle of the year, especially considering the logistics of securing residence. Nevertheless, we understand the reviewer’s concern and have added this point as a potential limitation in the discussion section our paper.

“Apart from that, we recognize that we did not account for physical activity as a covariate, and did not account for potential changes in living arrangement between the two assessment time points. As such, our findings should be interpreted in light of these limitations.”

4. Critical re-evaluation of the statistical analysis is warranted. The SDs are larger than the means in every case for the change data in table 2. Though a distribution-free test was used for comparison, it is still difficult to interpret the overall changes. Graphical representation of individual data points may be more informative. It is not clear how the distributions deviate from normality (e.g. skewed by weight loss, or excessive individual weight gain, etc)? Use of means for central tendency in skewed data is inappropriate. In the final discussion, only the group means are used to summarize support for the conclusions, but the individual changes are more relevant.

We thank the reviewer for the comments. The reviewer brings up several points and we have addressed each one of them individually below. 

A. The SDs are larger than the means in every case for the change data in table 2.

We do not think this is necessarily a problem. SDs are a measure of data spread or distribution. As such, the SDs in our table simply reflect the distribution of our data. We do agree that the precision of our data is not high due to our modest sample size and we have acknowledged the limitation of our sample size in the discussion section of our paper. However, apart from the fact that the large SDs reflect relatively lower precision in results, we do not think that the SDs being larger than the mean is an issue. 

B. Though a distribution-free test was used for comparison, it is still difficult to interpret the overall changes. 

In this case, we included tables that display the overall trends in our sample between the two time points, with a dedicated column indicating the magnitude and direction of change observed across all investigated traits. We believe that the tabulated data provides sufficient information for readers to interpret the overall changes in relation to the statistical results provided. Nevertheless, in order to improve clarity, we have updated all our tables and texts alike to include the 95% confidence intervals for all values of change, as we believe that a confidence interval would be more reflective of the spectrum of change observed in the sample for each of the investigated traits. Additionally, for readers who are interested, we have further included supplementary graphs (box plots and histograms) that display the distribution of data for all investigated traits across the three types of living arrangements explored in the study. 

C. Graphical representation of individual data points may be more informative.

We thank the reviewer for the comment. As mentioned above, we have now added supplementary graphs, particularly box plots and histograms, which display the distribution of data. Please see the ‘supplementary information’ document. 

D. It is not clear how the distributions deviate from normality (e.g. skewed by weight loss, or excessive individual weight gain, etc)?

Thank you for the comment. As discussed above, we have now specifically included histograms that illustrate the distribution for the change observed among participants in each of the living arrangement groups for all investigated traits. 

E. Use of means for central tendency in skewed data is inappropriate.

The reviewer definitely brings up a valid point. We agree with the reviewer that the use of means for central tendency in skewed data is inappropriate. Notably, in our case, given that we studied multiple traits (i.e. BW, BMI, WC, HC, and WHR) at multiple time points (i.e. beginning of the year, end of year, & change between the two time points), we examined the distribution of all the these traits at all relevant time points. Inevitably, some of that data is normally distributed while other parts are not. Particularly, in this case, the data pertaining to change (which is the primary focus of the paper) is not skewed for most traits. Hence, the use of means and SD in those cases is appropriate. 

With respect to the data that is skewed, we agree that the use of mean is not necessarily appropriate. However, given all the variation among the 5 different outcome variables at each time point, we decided to report all data as mean (SD) so that the data is presented consistently throughout the paper and is not confusing for the average reader. If we do change it, it would mean that we would have to have to switch back and forth between median and mean values in the tables and also in our results and discussion sections of our paper, which we feel would directly affect the readability and understanding of the content for the reader. Furthermore, to accommodate this, as per the reviewer’s previous recommendation, we have included supplementary graphs for readers who may be interested in accessing information regarding the distribution of traits. In any case, the statistical tests/p-values are unaffected. We accounted for distribution in each of our statistical tests using various methods where necessary (i.e. transformation, use of non-parametric tests).

F. In the final discussion, only the group means are used to summarize support for the conclusions, but the individual changes are more relevant.

Thank you for the comment. Given that this is a cohort study (i.e. a population level study), we believe that the discussion of group trends is appropriate. This is common practice for population studies wherein the summary statistics used (e.g. mean differences, risk ratios, odds ratios etc.) are based on examination of traits at the group level rather than at the individual level. Nevertheless, we agree with the reviewer that readers may be interested in seeing how individual data points are generally distributed. As such, we have included supplementary graphs that display distribution of all traits.

---

## [Editor Report · Decision Letter 1]

21 Oct 2020

Effect of living arrangement on anthropometric traits in first-year university students from Canada : the GENEiUS study

PONE-D-20-20895R1

Dear Dr. Meyre,

We’re pleased to inform you that your manuscript has been judged scientifically suitable for publication and will be formally accepted for publication once it meets all outstanding technical requirements.

Kind regards,

Rasheed Ahmad, Ph.D.

Academic Editor

PLOS ONE
---

## [Editor Report · Acceptance letter]

26 Oct 2020

PONE-D-20-20895R1 

Effect of living arrangement on anthropometric traits in first-year university students from Canada: the GENEiUS study 

Dear Dr. Meyre:

I'm pleased to inform you that your manuscript has been deemed suitable for publication in PLOS ONE. Congratulations! Your manuscript is now with our production department. 

Kind regards, 

on behalf of

Dr. Rasheed Ahmad 

Academic Editor

PLOS ONE